# A low-cost PM$_{2.5}$ monitor for wildland fire smoke

Scott Kelleher[1], Casey Quinn[2] Daniel Miller-Lionberg[1], and John Volckens[1]

[1]Department of Mechanical Engineering, Colorado State University, Fort Collins, USA
[2]Department of Environmental and Radiological Health Sciences, Colorado State University, Fort Collins, USA

*Correspondence to*: John Volckens (john.volckens@colostate.edu)

**Abstract.**

Wildfires and prescribed fires produce emissions that degrade visibility and are harmful to human health. Smoke emissions and exposure monitoring is critical for public and environmental health protection; however, ground-level measurements of

smoke from wildfires and prescribed fires has proven difficult, as existing (validated) monitoring technologies are expensive, cumbersome, and generally require line power. Few ground-based measurements are made during fire events, which limits our ability to assess the environmental and human health impacts of wildland fire smoke.

The objective of this work was to develop and validate an Outdoor Aerosol Sampler (OAS) - a filter-based air sampler that

has been miniaturized, solar powered, and weatherproofed. This sampler was designed to overcome several of the technical challenges of wildland fire monitoring by being relatively inexpensive and solar powered. The sampler design objectives were achieved by leveraging low-cost electronic components, open-source programming platforms, and in-house fabrication methods. A direct-reading PM$_{2.5}$ sensor was selected and integrated with the OAS to provide time-resolved concentration data. Cellular communications established via Short Message Service (SMS) technology were utilized in transmitting online

sensor readings and controlling the sampling device remotely. A Monte Carlo simulation aided in the selection of battery and solar power necessary to independently power the OAS, while keeping cost and size to a minimum.

Thirteen OAS were deployed to monitor smoke concentrations downwind from a large prescribed fire. Aerosol mass concentrations were interpolated across the monitoring network to depict smoke concentration gradients in the vicinity of the

fire. Strong concentration gradients were observed (spatially and temporally) and likely present due to a combination of changing fire location and intensity, topographical features (e.g. mountain ridges), and diurnal weather patterns. Gravimetric filter measurements made by the OAS (when corrected for filter collection efficiency) showed relatively good agreement with measurements from an EPA federal equivalent monitor. However, the real-time optical sensor (Sharp GP2Y1023AU0F, Sharp Electronic Co.) within the OAS suffered from temperature dependence, drift, and imprecision.

## 1. Introduction

Wildfires and prescribed fires are the largest combined source of primary fine particulate matter ($PM_{2.5}$) emissions into the atmosphere (NEI, 2014). These emissions degrade visibility and contribute to human morbidity and premature mortality (Sakamoto et al., 2016; Silva et al., 2013). Human exposure to biomass burning emissions has been associated with respiratory outcomes such as asthma, bronchitis, and COPD (Atkinson et al., 2014; Cohen et al., 2005; Gan et al., 2013) and cardiovascular outcomes such as high blood pressure, stroke, and arrhythmia (Brook et al., 2004; Clark et al., 2013; Pope et al., 2004).

Fire regimes have changed during the last century due to changes in climate, land-management techniques, agricultural practices, and industrial development (Westerling et al., 2006). Over the past three decades, wildfires have increased in number, size, and severity (Alves et al., 2000; Miller et al., 2009). This upward trend of wildfire activity is predicted to persist in coming years (Flannigan et al., 2000), meaning biomass burning will have an even greater impact on public and ecosystem health in the future (Spracklen et al., 2009; Yue et al., 2013). One method of wildfire mitigation is prescribed burning, a technique that has increased substantially in recent years.

The Interim Air Quality Policy on Wildland and Prescribed Fires was written by the EPA in 1998 to preserve public health and wellbeing by mitigating air quality impacts from prescribed fires (EPA, 1998). Evaluating the effectiveness of smoke mitigation techniques is challenging, however, because emission and exposure monitoring data are sparse. Conventional instruments for monitoring wildfire smoke are expensive, costing $10,000 to $30,000 per unit (Strand et al., 2011). These instruments are large and typically require line power; thus, sampling locations are often limited to areas equipped with utility service and accessible by motor vehicle. Thus, few measurements are made during most fire events, which often results in an incomplete representation of the fire's impact on local air quality (Hardy et al., 2001).

Satellite observations of air quality can be used to address ground-based monitoring gaps. Moderate resolution imaging spectroradiometer (MODIS) instrumentation, aboard the Terra and Aqua satellites, yield daily aerosol optical depth (AOD) measurements worldwide. AOD is an integrated extinction of light from the total mass of aerosol present in a vertical column of the atmosphere; thus, AOD includes total aerosol mass at all elevations. Satellite-based aerosol measurements, however, still lack precision and fine spatial resolution and do not quantify air quality specifically at ground level (Lassman et al., 2017). As a result, a need still exists for spatially-resolved measurements of surface air quality in the vicinity of fires.

The objective of this work was to develop and validate a field-deployable, low-cost (under $500) $PM_{2.5}$ sampler that can run autonomously with no external power. The sampler was intended to be remotely programmable and encapsulated in a lightweight, hardened enclosure. Specific design and performance objectives for the unit were: weigh less than 1000 grams (2.2 lbs), fit within a 3000 cm$^3$ (183 in$^3$) volume for ease of shipping, weatherproof, powered by solar and rechargeable battery,

capable of providing both online (via light scattering) and time integrated (via size-selective sampling onto a filter) measurements of $PM_{2.5}$, and capable of 1-2 week deployments. Following OAS development, a network of these low-cost

$PM_{2.5}$ monitors was deployed downwind from a prescribed fire to evaluate this device as a smoke-monitoring tool.

## 2. Methods

### 2.1 UPAS Technology

The Outdoor Aerosol Sampler (OAS) was based upon the Ultrasonic Personal Aerosol Sampler (UPAS) described previously (Volckens et al., 2017). The original UPAS was designed as a wearable device to estimate personal exposure to

$PM_{2.5}$ across a 24-hour period. A key feature of this sampler is a piezoelectric pump (which operates at ultrasonic frequency) that provides reductions in size, cost, and power relative to common diaphragm or rotary-vane air movers. The UPAS weighs 190 grams and has a bill-of-materials of approximately \$300. The sampler contains a size-selective cyclone inlet for $PM_{2.5}$, a 37-mm air sampling filter, rechargeable batteries, a suite of environmental sensors (location by GPS, temperature, pressure, relative humidity, acceleration, light, and a real-time clock), and a miniature mass-flow sensor. Sample duration

and volumetric flow-rate through the instrument (1-2 L/min) can be programmed into the device using a smartphone application via Bluetooth connectivity (available on iOS and Android platforms). In prior laboratory tests, the UPAS performed well compared to both an EPA $PM_{2.5}$ federal reference method (URG cyclone model URG-2000-30EGN-A; URG Corp., Chapel Hill, NC, USA) and a personal $PM_{2.5}$ sampler (PEM 761-203; SKC, Inc., Eighty Four, PA, USA).

### 2.2. OAS Development

The following additions and modifications were made to convert the UPAS from an indoor personal sampler into an outdoor area monitor: add a direct-reading $PM_{2.5}$ sensor, add cellular communications (Short Message Service, SMS), add battery capacity and solar charging, and harden the enclosure for all-weather operation. Additional modifications included developing a cartridge-style filter-holder (to simplify filter exchange in the field and to minimize sample contamination),

replacing the press-fit inlet cap with a threaded aluminum inlet, and modifying the housing to allow the device to be mounted into a weatherproof case.

Remote communications were accomplished by adding Short Message Service (SMS) technology, which allowed the OAS to be controlled via cell phone (or any device with internet access) and to report data back to a server. The built-in SMS

technology and predesignated communication protocol of a Particle Electron (Particle Industries Inc., San Francisco, CA) was utilized for this purpose. The Electron also features a microcontroller that was integrated into the UPAS circuitry, enabling communication among all components.

The online PM$_{2.5}$ sensor selected for this work was the Sharp GP2Y1023AU0F, which has been evaluated previously (Wang
et al., 2015). Wang's evaluation of the Sharp demonstrated a linear response with aerosol concentration change and less
dependency on atmospheric variables with respect to other low-cost sensors evaluated. This light-scattering detector was
envisioned to serve as a trigger mechanism, in addition to providing continuous measurements of PM concentration. The
sensor could prompt the OAS to begin sampling gravimetrically once a threshold of aerosol concentration was exceeded.
The prompt from the real-time sensor could also serve as an early warning alarm (transmitted via on-board SMS technology)
for an approaching smoke plume. Thus, the majority of OAS systems could remain idle until a smoke plume is detected,
which would help to conserve battery life.

A Pelican 1020 Micro Case was modified to enclose the OAS and to protect the unit from adverse weather. The mass-flow
sensor within the unit (Omron model DP6F) is dependent on air density, which can vary as a function of temperature,
humidity, and pressure. These variables were monitored in real-time using an atmospheric condition sensor (Bosch
BME280) and used to correct mass-flow readings (Volckens et al., 2017). The UPAS creates a small amount of heat during
operation as a by-product of battery discharge and pump work. Therefore, pump exhaust was routed through the case (and
out a series of small exit holes on the underside) to help maintain a temperature inside the case near ambient. At 2 L/min of
flow, approximately four air exchanges take place within the case each minute. Shielding provided by the solar cells reduces
OAS internal heating due to the absorption of solar radiation.

The OAS's two configurations are shown in Fig. 1.

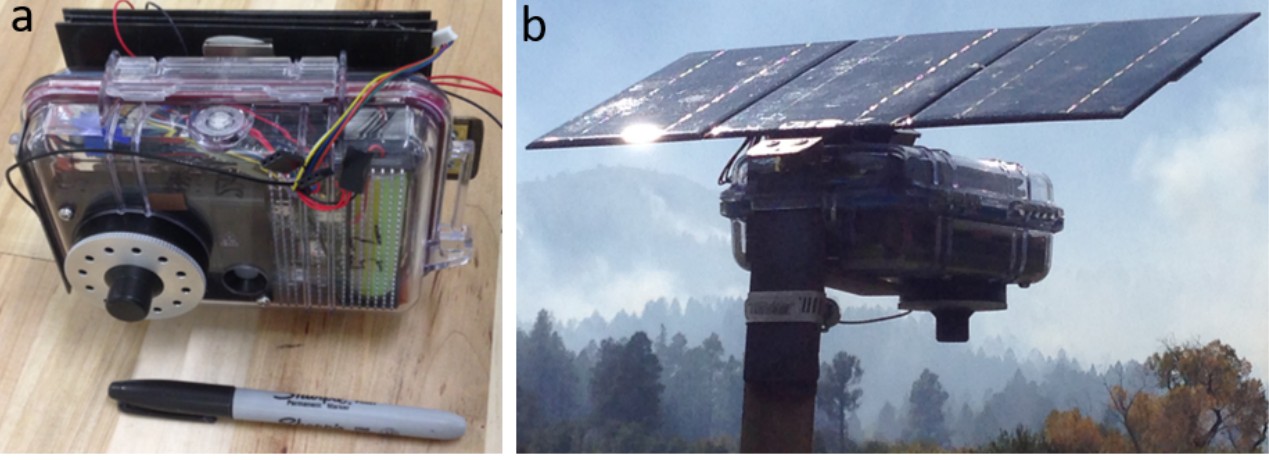

**Figure 1.** OAS sampler in (a) transportation configuration and (b) deployment configuration.

Table S1 lists the components added to the UPAS in the development of the OAS. The block diagram in Fig. 2 depicts the basic components of the UPAS (colored grey) and the additional components added to create the OAS (colored blue).

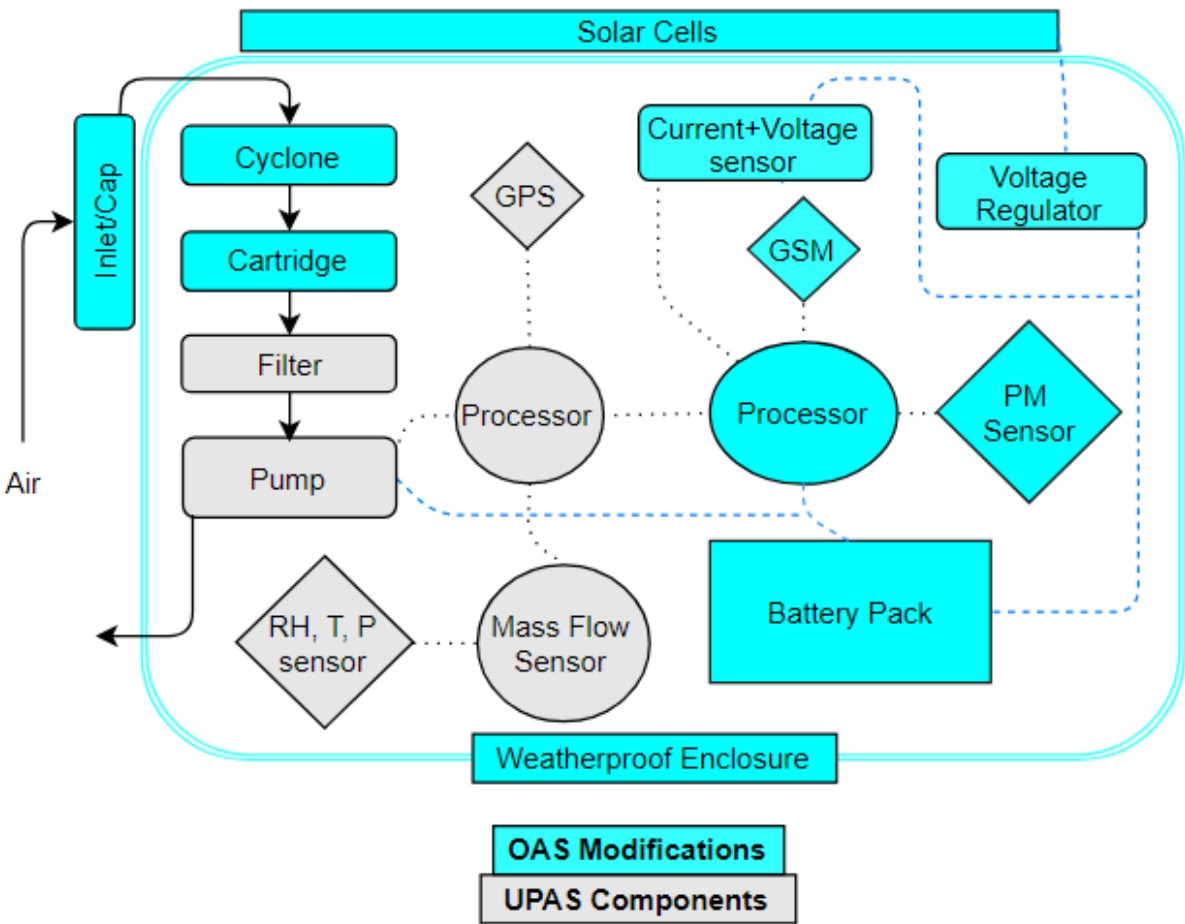

**Figure 2.** Block diagram of UPAS technology with component integration to form the OAS: GSM (Global System for Mobile Communications), PM (particulate matter), GPS (Global Positioning System)

### 2.3 Power System Design

Monte Carlo simulations were conducted to inform the selection of OAS solar and battery power. These simulations were designed to predict the probability of OAS power failure as a function of sampling duration (days), time of year (month), available solar irradiance, solar cell size, and battery capacity. Solar irradiance, the Monte Carlo sampled input variable, is defined as the daily average of observed solar irradiance attuned for solar cell size ($0.014m^2$ each). Solar irradiance data were obtained from the Christman Field weather station in Fort Collins, CO (CSU, 2016).

The simulation accounted for the following operational parameters: useable battery capacity, stationary solar conversion
efficiency, temperature effects on battery capacity, charging circuit efficiency, and average OAS power consumption at 2
L/min of sample flow. Power consumption also varies depending on filter type and filter loading, but for these simulations a
power consumption rate of 0.7 Watts was assumed (the approximate OAS power draw at 2 L/min of flow through a 37mm
Pallflex T60A20 Fiberfilm filter). One thousand iterations of 14-day sampling periods, for each month of the year, were
simulated to calculate runtime (in days) for each iteration. The probability of power failure (for a series of consecutive
sampling days across a particular month) is equal to the total number of failures specific to that day divided by the number of
iterations simulated. This calculation was repeated to estimate a failure probability for each of the 14 consecutive days that
were simulated across each month of the year. Monte Carlo simulations were run for varying numbers of solar panels (1-3
panels) and battery cells (2-5 cells).

The solar cell arrangement was designed to be collapsible to maintain a slender profile for easy transportation and shipping.
A magnetically coupled bracket that is adjustable for optimum zenith-angle holds the solar cells rigidly in place while in
deployment and transportation configuration (Fig. 1). A voltage regulator was added to the OAS battery charge controller to
condition electricity from the solar cells to 5 volts DC.

**2.4 Prescribed Fire Sampling**

Thirteen OAS units were arrayed in the vicinity of a 6000-acre prescribed fire (known as the Pargin Mountain fire), with
assistance from the Colorado Department of Public Health and Environment (CDPHE) and the US Forest Service (USFS).
The fire took place 14 km east of Bayfield Colorado from September 8[th] through September 17[th] 2016. The OAS network was
deployed prior to fire ignition, typically downwind and downslope from expected fire regions, following input from USFS
overseers. Other considerations for sampler placement included: cooperation from land owners, non-obtrusive to fire
operations, potential for livestock interference, and ease of access. Each OAS was placed on a tripod at a height of 1 m and at
a minimum of 60 m from the nearest road to avoid the influence of road dust emissions.

A map of sampler locations and the area burned is shown in Fig. 3. The USFS monitored air quality during the prescribed burn
by placing instruments at location 9 (E-SAMPLER, Met One Instruments, Grants Pass, OR) and location 1 (E-BAM, Met One
Instruments, Grants Pass, OR). Two OAS were co-located with each USFS monitor at these locations. For the duration of the
fire, each OAS was programmed to sample $PM_{2.5}$ for 24 hours onto 37mm Tisch PTFE filters (model SF17382) at a flowrate
of 2 L/min. The experimental design originally called for the use of Pallflex Fiberfilm T60A20 filters but these were

discontinued by the manufacturer; Tisch PTFE filters were selected as an alternate. Flow through the OAS was checked pre-
and post-sampling using a soap-bubble calibrator (A.P. Buck, Inc. Orlando, FL, USA).

Solar energy conversion efficiency was evaluated for each OAS and across all sampling periods. Data from the voltage/current
sensor on the OAS circuit board were used to determine the ratio of solar energy delivered to OAS batteries relative to available
solar irradiance. Hourly irradiance measurements were provided by a weather station (PRAWS 5) located on Pargin Mountain
during the month of September 2016.

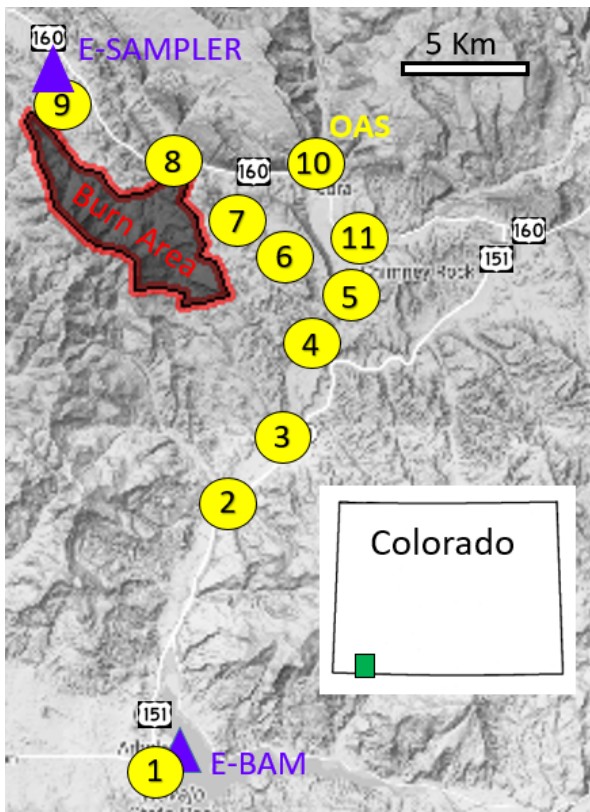

**Figure 3.** Location of monitoring equipment with respect to prescribed fire. OAS (yellow circles), US Forest Service equipment (blue triangles), prescribed fire (shaded black area with red outline). Map elements created using ggplot (Kahle and Wickham, 2013).


**Post-fire Performance Evaluation**

Following the Pargin fire deployment, we verified the accuracy and precision of the OAS with respect to time-integrated $PM_{2.5}$
measurements. In the laboratory, ten OAS units were arrayed with three $PM_{2.5}$ impactor samplers (PEM $PM_{2.5}$ 2 L/min, SKC
Inc.) in a 0.75 $m^3$ aerosol chamber to verify OAS accuracy and precision relative to a commercially-available $PM_{2.5}$ sampler
operating at similar flow rate. Sodium chloride was used as the test aerosol following the protocol described in Volckens et al.

(2017). Additionally, we evaluated OAS precision through a series of outdoor deployments whereby two OAS devices were co-located outdoors to sample ambient air concentrations for 48hr in Fort Collins, CO (n=23 paired deployments). From these tests, instrument precision was estimated from the coefficient of variation among co-located instruments and also as a mean absolute difference in measured concentration ($\mu g/m^3$) between paired instruments; OAS accuracy was estimated by calculating the average percent difference in measured concentration between the OAS and PEM samplers.

## 2.5 Sample and Data Analyses

Filters were contained in individual filter-keepers, inside sealed plastic bags, for both transportation and storage. Filters were placed in an equilibrium chamber for at least 12 hours before pre- and post-weighing and discharged on a polonium-210 strip for a minimum of 15 seconds prior to weighing on an analytic microbalance (Mettler Toledo XS3DU; ± 1 µg). Three readings were averaged together to determine each filter weight and field blanks were carried for all deployment days. Following post-weigh analysis, the filters were placed in filter keepers (SKC 225-8303) and sealed in air tight bags and stored at -20 °C.

Descriptive statistics were calculated for all mass concentration data, including identification of outliers, which were primarily cases when the OAS was explicitly known to have malfunctioned (stopped sampling, underflowed, etc). For OAS performance comparison with respect to USFS equipment, measurements were considered valid if the sampler spanned more than 75 percent runtime and flow remained within 20 percent of desired control. Limit of detection for gravimetric measurements was defined as the average blank mass gain plus three times the standard deviation of the change in blank mass. Limit of quantification was defined similarly but using five times the standard deviation in blank mass change.

Data analyses were conducted using Excel 2016 (Microsoft Corp., Redmond, WA, USA), Matlab 2015 (The MathWorks Inc., Natick, MA) and R 3.3.2 (R Core Team, Vienna, Austria). Spatial interpolation prescribed fire sampling results was based on ordinary kriging methods and plotted using gstat in R (Benedikt et al., 2016; Pebesma, 2004). Model interpolations were constrained to an area (search radius) no more than 3km from a given sampler location. Performance evaluation of the OAS relative to E-BAM utilized an errors-in-variables model (Deming regression) to estimate a linear fit between methods.

## 3. Results and Discussion

Several key modifications helped streamline the use of the OAS in the field. The replaceable filter cartridge (Figure S5b) eliminated the need for direct filter handling in the field (during change outs), which reduced the risk of contamination and also aided in sample transport. A threaded aluminum inlet cap (Fig. S5) sealed the filter cartridge in place and provided a rough inlet to protect against intrusion by small insects. The added costs (bill of materials) to convert the UPAS into the OAS totalled $183 for a single unit (Table S1).

### 3.1 Power

Power failure probabilities (representing the chance the OAS will experience power failure before the conclusion of a given number of consecutive sampling days) are shown Fig. 4 for the month of April. These results demonstrate the trade-off

between run duration and the quantity of solar panels and battery cells inside the OAS (Fig. 4). Based on these simulations, a final design consisting of three solar panels (0.042m$^2$ total) and five battery cells (totalling 54 watt-hours of capacity) was chosen. This configuration maximized OAS run duration while also meeting specified design criteria for instrument cost, size, and weight.

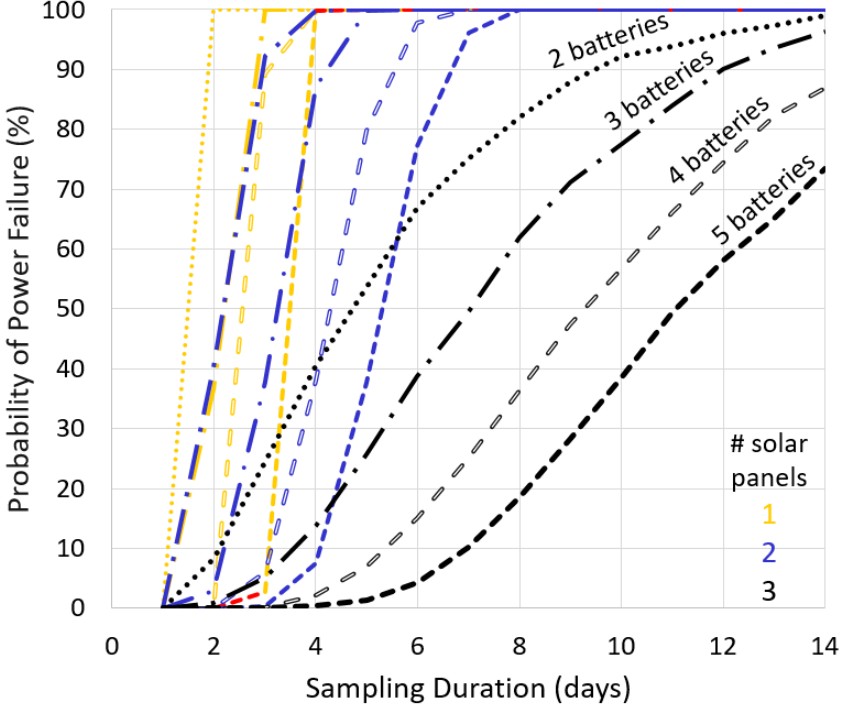

**Figure 4.** Probability of OAS power failure evaluated for various power designs (number of solar and battery cells) as a function of the number of continuous sampling days. Colors represent the number of solar panels (0.014m$^2$ each) and line type represents number of Li-ion batteries (10.78 W-hrs. each) included.

Power failure probabilities for the final OAS design are shown in Fig. S1 for six months of the year. The simulation results suggest that the OAS can achieve three full days of continuous sampling during late fall and winter, greater than four days in Spring months, and a full week of continuous sampling in Summer. The internal battery, when fully charged, allows for two full days of continuous sampling for all months of the year, regardless of the availability of solar power.

The Monte Carlo simulation was based on data from Fort Collins, Colorado between 2011 and 2015. Weather patterns, a large driver of solar irradiance available, are expected to vary by region. Thus, these simulation results are not generalizable beyond the Colorado Front Range. Further, the simulation selected random days within the specified month (i.e., blocks of consecutive days were not sampled) for any of the 4 years. Random selection of days can attenuate the effect of large weather systems, which may also impact OAS runtime.


### 3.2 Prescribed Fire

An early morning photo (Fig. 5) taken September 18, 2016 from Chimney Rock National Monument facing west shows the location of samplers in the OAS network during the Pargin burn. Smoke from the smouldering fire (red arrows) is observed down slope in the valley bottoms. An image captured on the morning of Sept 17$^{th}$ from a relay station (Fig. 6), 2.4 kilometers

northwest of location 9 facing east, depicts the OAS network from a second view point.

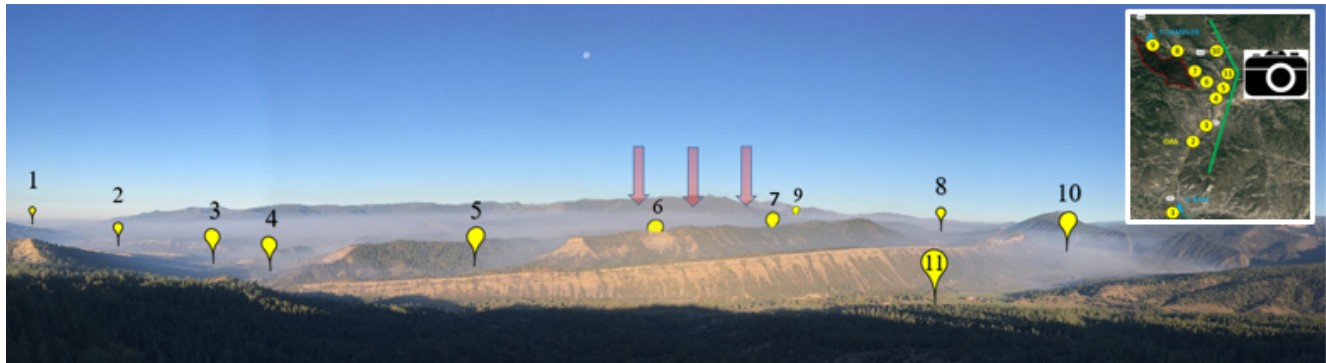

**Figure 5.** View from Chimney Rock, facing west on the morning of Sept. 18, 2016 when smoke is visible in several valleys. (Photo courtesy of Columbine Wildfire Management). OAS locations depicted by yellow markers. Visible smoke is observed around several

OAS while other locations appear to be smoke free. Red arrows indicate location of prescribed fire operations.

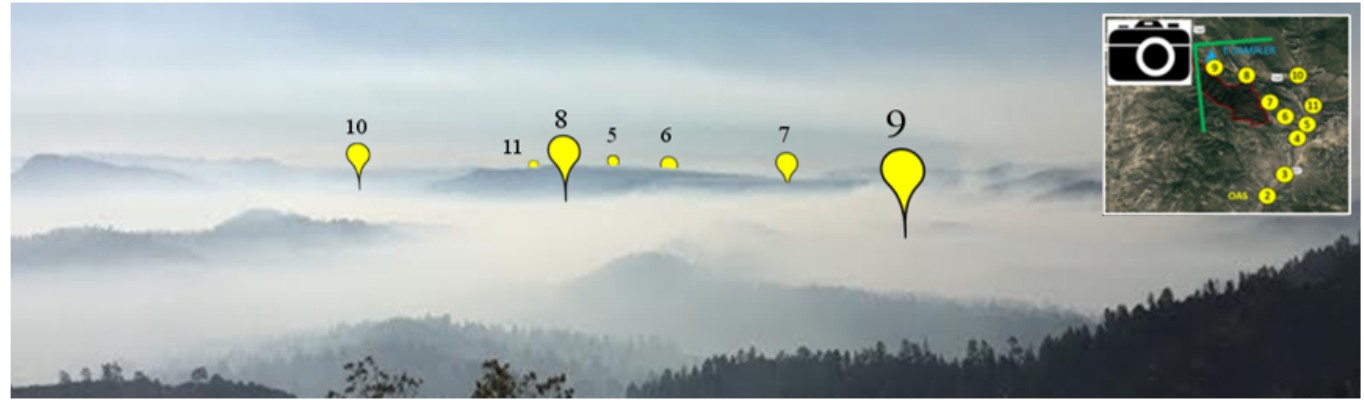

**Figure 6.** Smoke covering valley floors caused by an inversion on the morning of Sept 17[th], 2016. Photo taken from relay station 2.4 km northwest of Vance Ranch (location 9) facing east. (Photo courtesy of Columbine Wildfire Management)


The well characterized Pallflex Fiberfilm line of filters (originally intended for use within the OAS) was discontinued during this project; an alternative filter (Tisch PTFE) was selected. Tisch PTFE filters were selected because they exhibit a relatively low pressure drop and are comprised of hydrophobic polytetrafluoroethylene, which is less susceptible to organic vapor adsorption artifacts than other fibrous filter materials (Kirchstetter et al., 2001; Mader and Pankow, 2001). Prescribed

fire sampling results suggested a non-ideal collection efficiency for accumulation-mode aerosol using the Tisch PTFE filters. Subsequent laboratory tests, using a previously described protocol (Cardello et al., 2002), confirmed the relatively low collection efficiency of these filters (results shown in Fig. S2). The estimated mass collection efficiency of these filters was 66.7% (see supplemental material for a description of the method to evaluate filter collection efficiency), assuming a size distribution for an unaged biomass burning aerosol (Sakamoto et al., 2016). Mass concentration data reported here have been

corrected for filter collection efficiency.

A total of 61 OAS deployments were made over the nine-day prescribed fire. Seventeen of the sixty-one deployments failed to complete an intended measurement. Approximately half of these failures (Fig. S6, n=7) were due to premature power failure, defined as depletion of the battery before the conclusion of a 24-hr sampling period. Analysis of filter pressure drop

data (collected on board each OAS) and filter mass accumulation revealed that these failures occurred in sampling locations where $PM_{2.5}$ concentrations were extremely high, often exceeding a 24-hr average level of 200 μg/m$^3$. Power consumed by the OAS is strongly dependent on filter loading, which is a function of the sampled aerosol mass concentration. High filter loadings create increasingly larger pressure drops across the OAS filter, forcing the pumps to work harder (and thus consuming more battery power) to maintain a flow rate of 2 L/min. In these situations, if the OAS sampled for at least 10hrs,

the measured mass concentrations were extrapolated out to a 24-hr average for reporting purposes (i.e., a 10hr mass

concentration was multiplied by 10/24 to extrapolate the measurement to a 24hr average). This method of extrapolation is conservative but serves to maintain a standard metric for comparison across all sampling locations and days; further, in all cases the extrapolated $PM_{2.5}$ concentrations still exceeded 100 $\mu g/m^3$ - indicating the presence of extremely high PM levels.

Data mapping and interpolation techniques (ordinary kriging) were used to investigate the spatial and temporal evolution of ground-level $PM_{2.5}$ concentrations from September 12[th] through September 18[th], 2016. Maps illustrating interpolated mass concentrations for September 10[th], 12[th], and 18[th] are shown in Fig. 7. Results from September 15[th], and 17[th] are shown in Fig. S3. Aerosol mass concentrations are colored to depict concentration gradients for the following ranges: green, concentrations at or below the National Ambient Air Quality Standards (NAAQS) annual $PM_{2.5}$ limit (12$\mu g/m^3$); yellow, concentrations
falling between the annual limit (12$\mu g/m^3$) and the NAAQS 24-hour $PM_{2.5}$ limit (35$\mu g/m^3$); orange, concentrations falling between the 24-hour NAAQS limit (35$\mu g/m^3$) and 100$\mu g/m^3$; and red, concentrations in excess of 100 $\mu g/m^3$. Average, 24-hr mass concentrations are labelled at each individual sampling site on each map. Wind speed and direction data during each 24-hour sampling period are illustrated by a wind rose located to the right of each map. These interpolated concentration maps depict aerosol concentrations near the fire ranging from less than 15$\mu g/m^3$ to over 500$\mu g/m^3$ across the sampling
campaign.

Factors that may affect sensor performance include but are not limited to changes in aerosol size and refractive index, ambient humidity, and ambient temperature. Biomass burning aerosols are known to span a range of particle sizes and refractive indices; these properties can also change over time due to aerosol processing in the atmosphere (Vakkari et al.,
2014). Increases in humidity may lead to overestimation of (dry) aerosol mass concentration due to water uptake by hygroscopic particles. An ambient relative humidity of 60% is considered a lower threshold for water uptake to begin affecting nephelometer response (Chakrabarti et al., 2004); this level was exceeded for 38% of the sampling time during the Pargin fire. However, relative humidity rarely exceeded 70% during this period (7% of the time). Published growth factors for biomass burning aerosol are relatively low at 70% humidity (Rissler et al., 2006), indicating that water uptake from
particle hygroscopicity (and, thus, sensor response) was probably not substantial during the Pargin fire. The effect of

temperature on sensor response can be manifested by influencing particle size via gas-particle partitioning and by affecting the sensitivity/response of the photodiode and photodetectors inside the sensor itself.  The latter effect is shown in Figure S4.

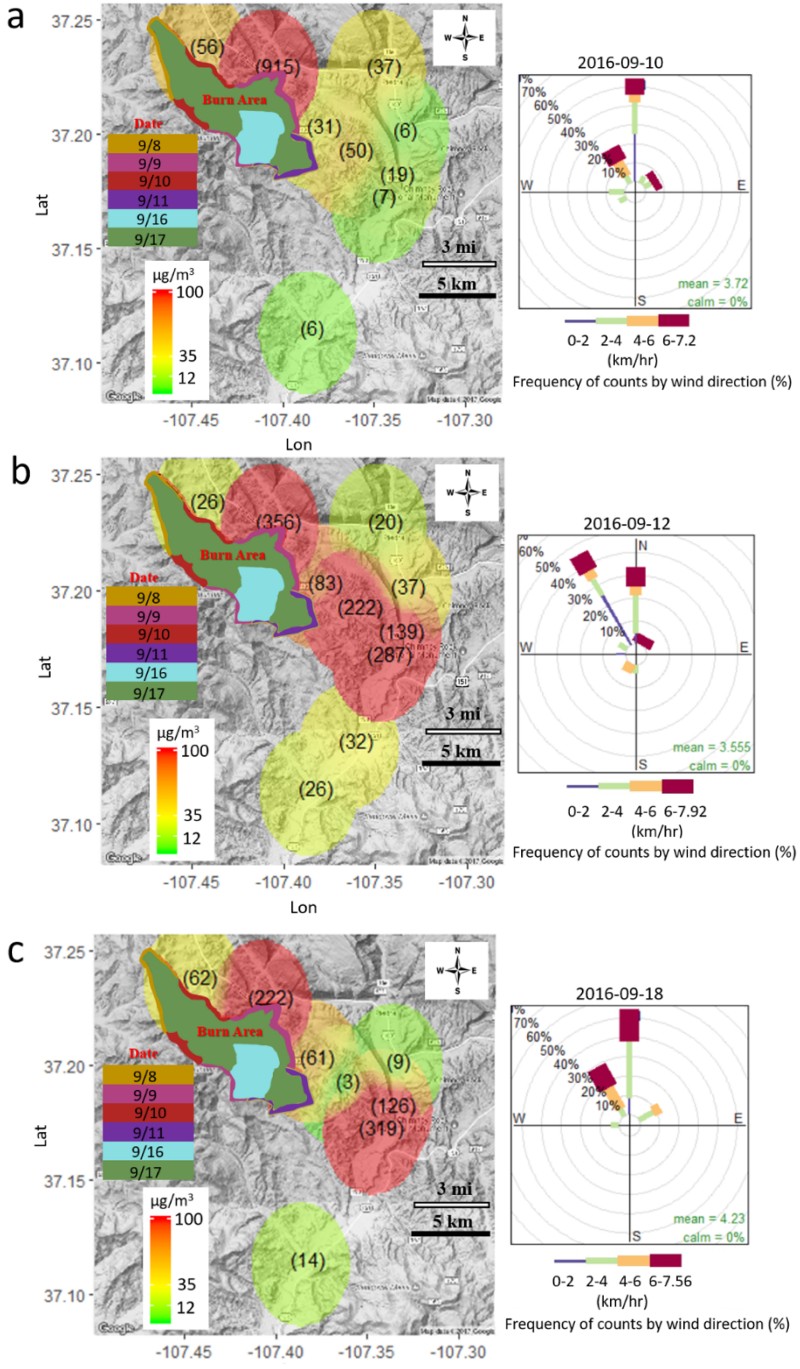

**Figure 7.** Maps illustrating spatial concentration gradients and the temporal evolution of fire emissions for (a) Sept 10[th], (b) Sept 12[th], and (c) Sept 18[th] of 2016. Numbers indicate 24-hour average mass concentration at each sampling site (e.g. '57' refers to a 24-hr mass concentration of 57 µg/m[3]). Dates represents the location and area burned each day with respect to the total prescribed fire.

A time series depicting variability of daily $PM_{2.5}$ concentrations measured across the OAS network is shown in Fig. 8. Measurements made by the US Forest Service air quality monitoring equipment (E-SAMPLER and E-BAM) are also shown as red and turquoise lines, respectively. The OAS network captured a wide range of $PM_{2.5}$ concentrations; this range was captured by the two USFS monitors on only two of the five deployment days. Thus, the OAS network provided a more spatially comprehensive assessment of smoke impact in the immediate vicinity of the prescribed burn. One key advantage of the OAS, in this regard, is that monitoring can take place in remote areas that lack line power (necessary to operate equipment like the E-BAM and E-SAMPLER). On several days the OAS network reported nearly 100-fold changes in 24-hr average $PM_{2.5}$ concentrations between sites that were separated by only a few kilometres, further demonstrating the high spatial variability in smoke emissions from the fire. On 9/10/2017, a sampler recorded a 24-hr average $PM_{2.5}$ concentration of 915 $\mu g/m^3$ – the highest reported value during the study. This measurement occurred during black-lining operations along the perimeter of the Pargin burn area and pertains to location 8 on Figure 3, which is the point nearest to the fire boundary. Measured $PM_{2.5}$ concentrations at this location were consistently high due to its close proximity to fire operations and also to meteorological conditions that favoured transport of emissions downwind and downslope.

The real time optical sensor (Sharp GP2Y1023AU0F) integrated with the OAS was determined to be unreliable for measuring PM in an outdoor setting. The sensor was affected by meteorological variables and inconsistent drift patterns, which precluded the use of this sensor as a trigger for the gravimetric sampler. The Sharp sensor's output voltage with respect to ambient temperature is displayed in Fig. S4, demonstrating a strong linear trend. A second issue with the Sharp sensor was baseline drift, which spanned as much as 50 $\mu g/m^3$ on some days (see Figure S7 for further detail). Unfortunately, the baseline drift was neither predictable nor correctable during the outdoor deployment (anecdotally, this drift was much less apparent during laboratory testing in a controlled environment). Collectively, these issues precluded the ability to establish a baseline concentration (without in-situ calibration) prior to each deployment, even with a co-located gravimetric reference sample. Further work on the practicality of the Sharp sensor for remote outdoor sampling is warranted.


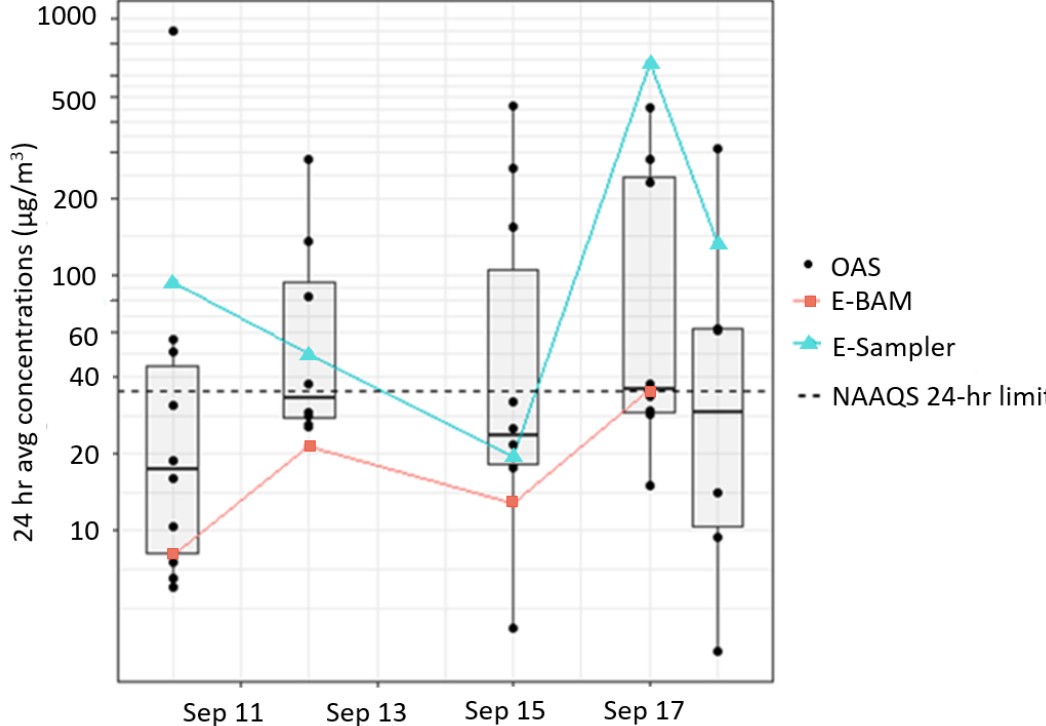

**Figure 8**. Prescribed fire summary of air quality at all locations for all dates sampled by both OAS and Forest Service equipment (E-BAM and E-SAMPLER). OAS concentrations displayed fit the following criteria: sample more than 75% of 24-hour sampling period and remain within 20% of desired flow control at all times.


A performance comparison between the OAS and E-BAM (co-located at Arboles Fire Station, location 1) is shown in Fig. 9. The E-BAM measures PM mass concentrations using Beta attenuation and has been shown to agree closely with FRM monitors (Trent, 2006). A Deming regression of the E-BAM and OAS yields a slope of 1.01 and an intercept of -5.9 µg/m$^3$. The intercept may be due to error in the estimated mass collection efficiency of the Tisch PTFE filters for biomass burning

aerosol. However, the agreement between the two instruments is still relatively good (R$^2$ = 0.92), despite the small sample size (n =7).

Solar power harvested by the OAS was compared to solar irradiance data for the duration of the prescribed burn sampling. On average, 6.7% of incident solar energy was converted into useful battery power by the OAS. Solar conversion efficiency

measured in the field was slightly less than the 7.5% efficiency input to the Monte Carlo simulation. On average, however, the solar circuit added an average of 11 hours runtime to the OAS during a given 24-hour period.

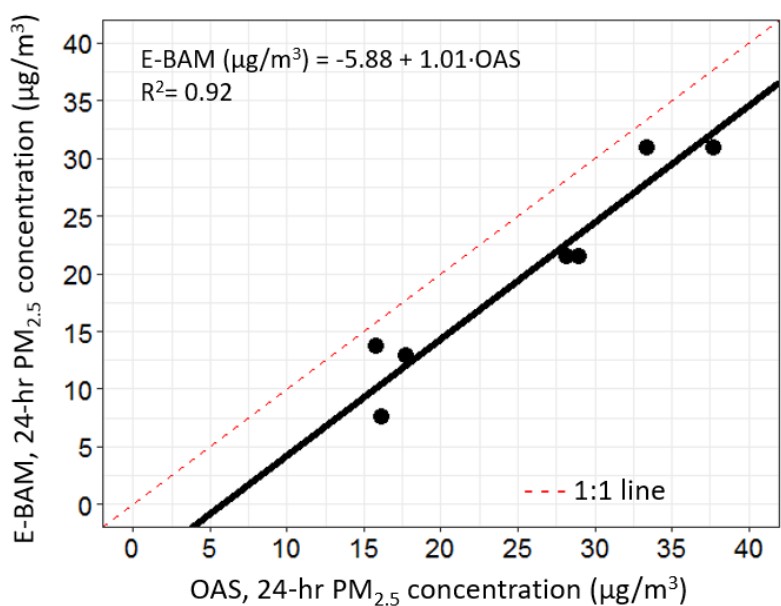

**Figure 9.** Performance of the Outdoor Aerosol Sampler relative to an E-BAM federal equivalent Monitor (meets US-EPA requirements for Class III designation for $PM_{2.5}$) at location 1.

For the post-fire validation experiments, the OAS and PEM samplers reported good agreement for sodium chloride aerosol measured in chamber tests (532±32 vs. 522±46 µg/m³, respectively); the average mass concentrations reported between instruments differed by only 2%. These results confirmed previous data reported by Volckens et al. (2017) that co-located the UPAS (the predecessor to the OAS) in a chamber with both PEM and FEM samplers. The coefficients of variation among co-located instruments were also similar: 8.9% for the PEMs and 7.9% for the OAS. For the outdoor deployments, the coefficient of variation among co-located OAS was 13%, which translated to an average difference in measured concentration of 1.4 µg/m³ at typical ambient PM2.5 concentrations (~8 µg/m³) in Fort Collins, CO. A tabular summary of these performance tests is provided in the online supplement.

## 3.3 Limitations and Future Work

The thirteen OAS samplers deployed in a network on the prescribed fire captured high concentration gradients resulting from smoke. Actual gradients, however, may have been even stronger than what was measured. One method of improving the spatial resolution of the network would be to deploy more OAS units. However, given the terrain features for the Pargin burn, only about 15 OAS units could be feasibly deployed by a single person in a 24-hr period.

The assumption of a fixed OAS power consumption did not allow the simulation to account for high filter loadings and the associated increased OAS power consumption. High aerosol concentrations (i.e., > 200 µg/m$^3$) also reduced the OAS runtime due to excessive filter loading rate. Future work should consider strategies to improve runtime when sampling extremely high aerosol concentrations, such as the use of intermittent sampling flow via duty cycle.


A hindrance of the remote communications method used is the limited availability of the Particle hosted service. The service is only available while the Particle Electron is online, resulting in increased power consumption if communication is to be maintained at all times. Another issue is the execution frequency permitted by Google Scripts. Google Scripts is a free service; however, execution frequency limits data collection to once per hour. A possible solution addressing the limited availability

of the Particle web page would be the use of an interrupt queue. This prompt would significantly reduce server time and power consumption. A personal server designed for OAS communication would alleviate issues associated with data collection frequency and simplify data archiving.

The Sharp sensor suffered from unpredictable drift issues, rendering the real-time measurements unreliable. Although post-

sampling calibration (i.e., normalizing the sensor data to the 24-hr filter mass concentration) would alleviate some of this error, the baseline drift issue (Figure S7) would still produce a substantial bias in reported PM concentrations. Possible OAS improvements include replacing the real-time sensor (Sharp) with a more reliable PM sensor. Low cost PM$_{2.5}$ optical sensing technology is an active area of research and development (Crilley et al., 2017; Sousan et al., 2016); future iterations of the OAS technology should seek to improve this capability. An accurate, reliable low-cost sensor would enable the OAS to monitor

air quality while remaining in an idle state. Utilizing the low-cost sensor as a trigger mechanism (as originally intended) would allow an OAS network to serve as an early warning tool by detecting and tracking emissions in real time.

### 4. Conclusions

Reference instruments used to assess outdoor air quality tend to be expensive and bulky. This project developed and tested an Outdoor Aerosol Sampler (OAS) that is compact, weatherproof, battery powered, and designed to approach reference-quality

measurements of PM$_{2.5}$. The OAS achieved a relatively compact size (17 x 12 x 10 cm), a low weight (888 grams) and quiet operation. The inclusion of solar and additional battery power allowed the OAS to be successfully deployed for several days at a time. The integration of wireless remote communications provides control of the OAS from distant locations and data transmission in real-time. A durable weatherproof enclosure and solid mounting system allowed the OAS to be operated during all months of the year, through strong winds, rain, and snow.


Thirteen OAS were deployed around a large prescribed fire in southern Colorado to evaluate its effectiveness as a smoke monitoring tool. The OAS network provided spatially resolved measurements in regions where sampling with current state-

of-the-art equipment was not possible. Strong concentration gradients were observed and likely present due to topographical features (e.g. mountain ridges) and diurnal weather patterns. At extremely high concentrations (i.e., 24-hr $PM_{2.5} > 200$

$\mu g/m^3$), the OAS units were prone to power failure due to overloading of the sampling filter.

The cost, independent power capability, and compactness of the OAS provide a practical means for more effective monitoring of smoke from a prescribed burn or wildfire event. A successful demonstration of a low-cost sensor network represents a first step towards providing burn managers, state and federal agencies, and concerned citizens with a better understanding of fire

smoke emissions and resulting exposures. The OAS is not limited to fire events only and may be used for many other applications of outdoor air quality monitoring. At nearly $1/20^{th}$ the cost of current state-of-the-art field monitoring equipment, the OAS may be deployed in higher quantities under the assumption of fixed fiscal resources. Air quality data at more locations has the potential to enhance the accuracy of exposure models, yielding a more comprehensive estimate of potential human and environmental health hazards from smoke.

**5. Data availability**

A .csv file of 24-hour concentrations and other sample data (start/stop times, locations, sampled air volumes, run times) from the Pargin fire and post-fire evaluation experiments is available as part of an online supplement.

**Acknowledgements**

The authors wish to thank the following individuals for their contributions to this work: Josh Smith for software and firmware

development; Nick Good, John Mehaffy, Christian L'Orange for assistance with field sampling and data analysis methods; John (Jay) Godson (USFS), Sarah Gallup (CDPHE), Pat McGraw (CDPHE), and Ken Helcoski (CDPHE) for their coordination with field sampling. This research was funded in part by a cooperative agreement from the Joint Fire Science Program (16-2-01-3).

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
