# Peer review of "A low-cost PM2.5 monitor for wildland fire smoke"

_Atmospheric Measurement Techniques, 2017_

## Referee Comment (RC1) · Anonymous Referee #1 · 31 Oct 2017

In general, the paper present new findings on use of lower cost technologies deployed during a fire event. The authors need to better present the results of the OAS sampler and expand on the discussion of the failed Sharp sensor as described in detailed comments below. Further, the authors need to expand on the characteristics known to impact sensor performance (e.g. type of wood burned, humidity, inversion vs. non inversion days, temperature, and wind direction) in the discussion and results section.

Specifically, the discussion on OAS results over 200 is confusing. Where were these located, how many out of the 61 sensors were affected by this issue, and further describe what you mean by extrapolating over 24-hours? Figure 8 clearly shows an outlier near 1000 (which needs discussion) and other values above 200. Also spend some time discussing the September 17th results (was this the only day of the inversion)? Figure 8 shows the reference value reading near the high point of ∼500 on the 17th. Can you create a similar figure that identifies each monitor? Was the highest recorded

value during the inversion the monitor closest to the reference? Besides collocation of OAS monitors at sites 1 & 9, describe the evaluation of precision & accuracy amongst the sensors before, during, and after the study.

Starting at line 279 - there is only brief discussion on temperature and drift, describe other met conditions affecting the Sharp sensor.

Future work could involve mobile monitoring with reference instruments to collocate sensors in the highest concentration environments. Also discuss whether a different low cost, real-time sensor with greater concentration ranges or known size ranges should be used.

---

## Referee Comment (RC2) · Anonymous Referee #2 · 25 Nov 2017

This paper by Kelleher describes the design of a low-cost and field portable PM2.5 monitor that would be of interest to many readers, especially related to remote sampling without access to mains power. The authors give a thorough description of the design, components of their monitor that would enable one to replicate their monitor as well discussion on the consideration that went into the design. The authors demonstrated the use of the monitor in the field, describe the use to spatial mapping the distribution of smoke from a prescribed burn in Colorado, and demonstrate that the monitor was able to successfully capture daily PM2.5 mass concentrations that compared well to reference instruments. The only downside to the paper is that I would have liked to have seen more discussion on the on-line optical sensor (Sharp) to monitor PM and the reasons why it failed. This will help the reader understand the limitations of the Sharp sensor and so avoid similar problems. In this reviewer opinion, the manuscript falls within the scope of AMT and should be considered for publication after consideration

of the minor points below. Specific comments 1. Page 4, line 95: Perhaps the authors could outline why they chose the Sharp optical sensor over the myriad of other low-cost particle sensors available commercially. 2. Section 2.4: what sampling height did you place the monitors? 3. Page 9, line 216: Perhaps you could compare the total battery life that was achieved during the measurements compared to the simulations? 4. Page 11, line 240: While I agree that you should correct for collection efficiency of the filter, it would help if the authors were a bit more explicit in how the correction was applied 5. Page 11, line 245: why would the high mass loading reduce battery power, was it because the pumps had to work harder than expected? 6. Figure 7: For each site on the maps you give an 'average' PM2.5 daily mass concentration. How is this an average concentration when each map is one day of measurements and there is presumably one OAS at each site? 7. Figure 7 and S3: what happed to data from the 11th, 13th, 14th and 16th Sept? Why is data from these days not included in the Supplement? 8. As mentioned previously, I think more discussion on the why the Sharp optical sensor failed would be useful. The authors mentioned baseline correction was attempted but give no details, what variables were tried but failed? Or to put it another way, how did you come to conclusion that the baseline was not correctable? As the strong linear trend with temperature suggested that it could be correctable. By giving details of what did not work is just as valuable for the reader, so that they can avoid similar issues with this sensor. Furthermore, have other people reported the same problems (e.g. Wang et al. 2015)? 9. Figure 9: What about the results from the second location were the OAs was co-located with reference instruments, did it perform just as well? 10. Page 16, line 333: there have been a few recent papers that have found reliable low-cost and small optical particle counters (e.g. Crilley et al.2017 & Sousan et al. 2016). I think the authors could reference these and other papers here with discussion on whether these other sensors may be useful and practical.

References Crilley, L. R., Shaw, M., Pound, R., Kramer, L. J., Price, R., Young, S. Lewis, A.C. and Pope, F. D. 2017. Evaluation of a low-cost optical particle counter (Alphaseense OPC-N2) for ambient air monitoring. Atmospheric Measurement Tech-

niques Discussion. In review. Sousan, S., Koehler, K., Hallett, L. and Peters, T.M., 2016. Evaluation of the Alphasense optical particle counter (OPC-N2) and the Grimm portable aerosol spectrometer (PAS-1.108). Aerosol Science and Technology, 50(12), pp.1352-1365.

---

## Author Comment (AC1) · 9 Jan 2018

We thank the reviewer for their insightful comments, which have allowed us to produce a stronger manuscript. Our responses to the general and specific comments are given below. Please note that line number references pertain to the "tracked changes" version of the revised manuscript.

Reviewer 1 general comments: In general, the paper present new findings on use of lower cost technologies deployed during a fire event. The authors need to better present the results of the OAS sampler and expand on the discussion of the failed Sharp sensor as described in detailed comments below.

Response: Please find our responses to the comments and clarifications detailed below, and details of the corresponding changes in the revised manuscript.

[Figure]

Reviewer 1 specific comments

1. Comment: The authors need to expand on the characteristics known to impact sensor performance (e.g. type of wood burned, humidity, inversion vs. non inversion days, temperature, and wind direction) in the discussion and results section.

Response: We agree that factors such as particle size distribution and refractive index, as manifested through changes in ambient humidity, or the type of wood burned, and/or atmospheric aerosol processing, would affect sensor response. We also agree that ambient temperature and humidity could affect sensor performance (at high humidity the optics can become obscured due to condensation). While inversions and wind direction may influence ambient aerosol concentrations, we do not see how these factors would affect sensor performance, per se.

We have revised lines 285-295 of the manuscript to: "Factors that may affect sensor performance include but are not limited to changes in aerosol size and refractive index, ambient humidity, and ambient temperature. Biomass burning aerosols are known to span a range of particle sizes and refractive indices; these properties can also change over time due to aerosol processing in the atmosphere (Vakkari et al., 2014). Increases in humidity may lead to overestimation of (dry) aerosol mass concentration due to water uptake by hygroscopic particles. An ambient relative humidity of 60% is considered a lower threshold for water uptake to begin affecting nephelometer response (Chakrabarti et al., 2004); this level was exceeded for 38% of the sampling time during the Pargin fire. However, relative humidity rarely exceeded 70% during this period (7% of the time). Published growth factors for biomass burning aerosol are relatively low at 70% humidity (Rissler et al., 2006), indicating that water uptake from particle hygroscopicity (and, thus, sensor response) was probably not substantial during the Pargin fire. The effect of temperature on sensor response can be manifested by influencing particle size via gas-particle partitioning and by affecting the sensitivity/response of the photodiode and photodetectors inside the sensor itself. The latter effect is shown in Figure S4."

2. Comment: Specifically, the discussion on OAS results over 200 is confusing. Where were these located, how many out of the 61 sensors were affected by this issue, and further describe what you mean by extrapolating over 24-hours?

Response: We apologize for the confusion here. Some of our monitors did not sample for the full 24hrs because of premature power failure (i.e., the battery ran out of charge). Most of these power failures were due to overloading of the air sampling filter under extremely high PM levels. We wanted to include these high PM events in Figure 7, so we decided to extrapolate any measurements that lasted at least 10 hours up to a 24hr average. We recognize that extrapolating a 10hr measurement to a 24hr average is very conservative (this assumption essentially combines a 10hr measurement with an additional 14hrs of zero values). However, most of these averages are still above 200 $\mu$g/m3 – which means the OAS was measuring an extremely high PM concentration!

Lines 261-271 of the revised manuscript have been updated to: "Approximately half of these failures (Fig. S6, n=7) were due to premature power failure, defined as depletion of the battery before the conclusion of a 24-hr sampling period. Analysis of filter pressure drop data (collected on board each OAS) and filter mass accumulation revealed that these failures occurred in sampling locations where PM2.5 concentrations were extremely high, often exceeding a 24-hr average level of 200 $\mu$g/m3. Power consumed by the OAS is strongly dependent on filter loading, which is a function of the sampled aerosol mass concentration. High filter loadings create greater than normal pressure drops across the OAS filter, forcing the pumps to work harder (and thus consuming more battery power) to maintain a flow rate of 2 L/min. In these situations, if the OAS sampled for at least 10hrs, the measured mass concentrations were extrapolated out to a 24-hr average for reporting purposes (i.e., a 10hr mass concentration was multiplied by 10/24 to extrapolate the measurement to a 24hr average). This method of extrapolation is highly conservative but serves to maintain a standard metric for comparison across all sampling locations and days; further, in all cases the extrapolated PM2.5 concentrations still exceeded 100 $\mu$g/m3 - indicating the presence of extremely
high PM levels."

We recognize the value of identifying the locations of OAS failures on each sampling day. Therefore, we have revised Figure S6 to show monitor failures modes (and completed measurements) as a function of sampler location (as defined in Figure 3). We have also added these data (including run-times) to our supplementary data file.

3. Comment: Figure 8 clearly shows an outlier near 1000 (which needs discussion) and other values above 200.

Response: The extremely high concentration on Sept 10th captured by one of OAS is due to black lining operations taking place upslope and in the vicinity of the sampler. Lines 308-312 have been revised: "On 9/10/2017, a sampler recorded a 24-hr average PM2.5 concentration of 915 $\mu$g/m3 – the highest reported value during the study. This measurement occurred during black-lining operations along the perimeter of the Pargin burn area and pertains to location 8 on Figure 3, which is the point nearest to the fire boundary. Measured PM2.5 concentrations at this location were consistently high due to its close proximity to fire operations and also to meteorological conditions that favoured transport of emissions downwind and downslope."

4. Comment: Figure 8 shows the reference value reading near the high point of âĹij500 on the 17th. Can you create a similar figure that identifies each monitor? Was the highest recorded value during the inversion the monitor closest to the reference?

Response: Shown in Figure 7 and S3 are individual sampler locations and measured concentration data for each sampling day. A few data points shown in Figures 7 and S3 have been extrapolated due to premature power failure (see response to comment 2 above) – these results are not reported in Figure 8, where only data for samplers that completed at least 75% of sampling period are compared with data from the reference monitors.

In regard to the last question, (Was the highest recorded value during the inversion

the monitor closest to the reference?) the answer is 'no'. The E-sampler reference instrument was positioned at location 9 (see Figure 3). Referring to Figure S3.B, the OAS at location 9 recorded a 24 hour concentration of 287 $\mu$g/m3 on 17 September 2016. The OAS adjacent to this monitor at location 8 recorded the highest value on that day.

5. Comment: Besides collocation of OAS monitors at sites 1 & 9, describe the evaluation of precision & accuracy amongst the sensors before, during, and after the study.

Response: We did conduct additional evaluation of OAS units following the Pargin fire deployment. These tests were conducted to verify that the OAS units were performing as expected. We now describe these tests in the Methods/Results sections:

Added to Methods (lines 174-184):

"Following the Pargin fire deployment, we verified the accuracy and precision of the OAS with respect to time-integrated PM2.5 measurements. In the laboratory, ten OAS units were arrayed with three PM2.5 impactor samplers (PEM PM2.5 2 L/min, SKC Inc) in a 0.75m3 aerosol chamber to verify OAS accuracy and precision relative to a commercially-available PM2.5 sampler operating at similar flow rate. Sodium chloride was used as the test aerosol following the protocol described in Volckens et al. (2017). Additionally, we evaluated OAS precision through a series of outdoor deployments whereby two OAS devices were co-located outdoors to sample ambient air concentrations for 48hr in Fort Collins, CO (n=23 paired deployments). From these tests, instrument precision was estimated from the coefficient of variation among co-located instruments and also as a mean absolute difference in measured concentration ($\mu$g/m3) between paired instruments; OAS accuracy was estimated by calculating the average percent difference in measured concentration between the OAS and PEM samplers."

Added to Results and Discussion (lines 348-355):

"For the post-fire validation experiments, the OAS and PEM samplers reported good

agreement for sodium chloride aerosol measured in chamber tests (532+/-32 vs. 522+/-46 $\mu$g/m3, respectively); the average mass concentrations reported between instruments differed by only 2%. These results confirmed previous data reported by Volckens et al. (2017) that co-located the UPAS (the predecessor to the OAS) in a chamber with both PEM and FEM samplers. The coefficients of variation among co-located instruments were also similar: 8.9% for the PEMs and 7.9% for the OAS. For the outdoor deployments, the coefficient of variation among co-located OAS was 13%, which translated to an average difference in measured concentration of 1.4 $\mu$g/m3 at typical ambient PM2.5 concentrations ($\sim$8 $\mu$g/m3) in Fort Collins, CO. A tabular summary of these performance tests is provided in the online supplement."

6. Comment: Starting at line 279 - there is only brief discussion on temperature and drift, describe other met conditions affecting the Sharp sensor.

Response: Please see our response to Reviewer #2, Comment 8, where we further describe the types of drift observed when using the Sharp sensor outdoors. A discussion of these phenomena (and an accompanying figure) has been added to the online supplement provided with the manuscript.

The performance of the Sharp sensor regarding its sensitivity to limit of detection, dependence on compositions, sensitivity to particle size, relative humidity influence, and temperature influence have been previously reported by Wang et al.:

Wang, Y., et al., Laboratory Evaluation and Calibration of Three Low-Cost Particle Sensors for Particulate Matter Measurement. Aerosol Science and Technology, 2015. 49(11): p. 1063-1077.

7. Comment: Future work could involve mobile monitoring with reference instruments to co-locate sensors in the highest concentration environments. Also discuss whether a different low cost, real-time sensor with greater concentration ranges or known size ranges should be used.

[Figure]

Response: We agree that selection of an improved real-time aerosol sensor with improved performance should be the result of future work.

We note on lines 379-380:

"Low cost PM2.5 optical sensing technology is an active area of research and development; future iterations of the OAS technology should seek to improve this capability."

Please also note the supplement to this comment:
https://www.atmos-meas-tech-discuss.net/amt-2017-358/amt-2017-358-AC1-supplement.pdf

**Supplement:**

**Supplemental Material: A low-cost PM$_{2.5}$ monitor for wildland fire smoke**

Scott Kelleher[1], Casey Quinn[2] Daniel Miller-Lionberg[1], and John Volckens[1]

[1]Department of Mechanical Engineering, Colorado State University, Fort Collins, USA

[2]Department of Environmental and Radiological Health Sciences, Colorado State University, Fort Collins, USA

*Correspondence to*: John Volckens (john.volckens@colostate.edu)

**Table S1.** Components added to UPAS to form the OAS

| Component | Manufacturer | Part number | Cost |
|---|---|---|---|
| Polycrystalline Solar Cell | Banggood | 991137 | 3@$5 |
| Voltage Regulator | ProDCtoDC | 90462 | $5 |
| Microcontroller/SMS Module | Particle | Electron 3G | $59 |
| MicroSD card logger | Molex | 5031821852 | $7 |
| Battery (2800 mAh) | Anker | 7OSMS5-28N | 3@$14 |
| Temp, Pressure, RH sensor | Bosch Sensortec | BME280 | $10 |
| Current/Voltage Sensor | Texas Instruments | INA219 | $10 |
| Low-cost PM Sensor | Sharp | GP2Y1010AU0F | $8 |
| Sharp Sensor adapter | DFRobot | DFR0280 | $4 |
| Weatherproof enclosure | Pelican | 1020 micro | $14 |
| Magnets | KJ Magnetics | BX08H1 | 7@1.3 |

Table S2 lists all input variables used in the simulation design. The amount of daily solar irradiance available (*S*), is the simulation's only Monte Carlo sampled input.

**Table S2.** Power design model variables

| Variable | Term | Input (units) | Data Source |
|---|---|---|---|
| *R* | Rated Battery Capacity | 10.78 (Watt-hours) | Determined Empirically |
| *E* | Solar Circuit Efficiency | 7.50% | Determined Empirically |
| *N* | Battery Quantity | 5 (unit less) | Determined Empirically |
| *T* | Temperature | Monthly mean of daily low temperatures(ºC) | CSU Christman Weather Station |
| *P* | OAS Daily Power Consumption | 16.8 (Watt-hours) | Determined Empirically |
| *S* | Solar Irradiance Available | Monte Carlo sampled daily value (watt-hrs.) | |
| *V* | Useable Battery Capacity Percentage | 0.85 (fractional percentage) | Determined Empirically |
| *C* | Battery Capacity Temperature Correction | Equation 2 (unitless) | |

Useable battery capacity percentage (V) was determined to be 85% based on OAS circuit cutoff voltages. Fully charged battery capacity $B_0$, where the subscript refers to day '0', can be expressed using Eq. (1);

$$B_0 = R * N * C_i * V \qquad (1),$$

where R is rated battery capacity, $N$ is battery quantity, $C_i$ is the capacity correction from temperature (for month i), and V is useable battery capacity percentage. The Li-ion battery capacity correction factor, $C$, as a function of month, $i$, for an 18650b battery type, is described in Eq. (2);

$$C_i = (-0.0097T_i^2 + 0.8061T_i + 90)/100 \qquad (2),$$

where $T(\degree C)$ is the mean low temperature across a given month, i . Eq. (3) is used to calculate the runtime (in days) for each of the 1000 sampling missions each month;

$$Runtime\ (days) = \sum_{d=1}^{14}\begin{cases}0\ if\ (B_{d-1} - P + S * E) > 0\\1\ if\ (B_{d-1} - P + S * E) < 0\end{cases} \qquad (3),$$

where $B$ is battery capacity at the conclusion of day $d$, $P$ is daily OAS power consumption, $S$ is available solar irradiance and $E$ is solar energy conversion efficiency.

[Figure]

**Figure S1.** Monte Carlo simulation results showing OAS power failure probability for every other month of the year. Axes define number of continuous sampling days and probability of power failure. Colors represent selected months spanning four seasons.

Particle collection efficiency of the Tisch PTFE filters (Figure S2) was evaluated in an aerosol chamber; wood smoke was used to simulate prescribed fire aerosol. A scanning mobility particle sizer (SMPS, model 3082, TSI Inc., Shoreview, MN) was used to count particles in 110 discrete size ranges from 19 to 1000 nm. A set of repeated measures was made with a Tisch PTFE filter inline and then removed (alternating the order between sets) for four test filters, three sets per filter. Additional chamber air (1.4 L/min) was metered through the filter to make up the

difference between the intended OAS flowrate (2 L/min) and the flow into the SMPS, which was nominally 0.6 L/min.

Filter collection efficiency for each particle size range was determined using Eq. (6);

$$filter\ collection\ efficiency_i = \eta_i = 1 - \frac{N_{i,on}}{N_{i,off}} \qquad (6),$$

where $N_{i,on}$ is the particle count measured by the SMPS with filter on, $N_{i,off}$ is the particle count measured by the SMPS with filter off and i represents the midpoint of each particle size range. Mass collection efficiency of the Tisch filter was estimated for prescribed fire aerosol using an aerosol size distribution specific to wildland fire. This distribution was modeled from a lognormal distribution having a count median diameter (CMD) of 70 nm and geometric standard deviation ($\sigma_g$) of 1.7 (Sakamoto et al., 2016). The mass of a single particle in each particle size range ($m_{p,i}$) was calculated using Eq. (7),

$$m_{p,i} = \frac{\pi}{6}d_i^3\,\rho \qquad (7),$$

where $d_i$ is the median particle diameter for each size range, i, and $\rho$ is particle density. The mass in each particle size range ($M_i$) can be calculated using Eq. (8),

$$M_i = N_i\,m_{p,i} \qquad (8),$$

where $N_i$ is the number of particles present in size range $i$. The mass collection efficiency of the filter is determined by the ratio of particulate mass collected by the filter to total particulate mass. Percent mass collection efficiency is calculated using Eq. (9):

$$mass\ collection\ efficiency = \frac{\sum_{i=1}^{n} \eta_i\,M_i}{\sum_{i=1}^{n} M_i} * 100 \qquad (9),$$

where the numerator on the right hand side represents the particulate mass collected by the Tisch filter (determined experimentally) summed up over n size ranges and the denominator is the summation of particulate mass across equivalent size ranges for a hypothetical biomass burning aerosol (Sakamoto et al., 2016). Filter collection efficiency has been found to increase with filter loading (Soo et al., 2016). The mass collection efficiency of a clean Tisch filter was considered to be constant when correcting prescribed fire filter mass accumulated for collection efficiency.

The average collection efficiency is depicted in Figure S2 by a black line; grey shading represents ± 1 standard deviation. The red plot represents a hypothetical aerosol mass distribution produced by prescribed fire (derived from (Sakamoto et al., 2016)).

[Figure]

**Figure S2.** Collection efficiency of 37mm Tisch PTFE filters (2L/min flow) with respect to particle mobility diameter and mass distribution of particles (red). Primary vertical axis represents filter collection efficiency; secondary vertical axis represents particle size distributions (by mass). Horizontal axis is particle size.

[Figure]

**Figure S3.** Maps illustrating spatial concentration gradients and the temporal evolution of fire emissions for Sept 15[th], and 17[th], 2016. Numbers indicate 24-hour average mass concentration at each sampling site (e.g. (57) refers to a mass concentration of 57 µg/m³).

[Figure]

**Figure S4.** Sharp sensor (GP2Y1023AU0F) output correlation with ambient temperature.

[Figure]

**Figure S5.** a) UPAS sampler with threaded aluminium inlet in place b) threaded aluminium inlet, size-selective cyclone, and filter cartridge.

The run performance of each OAS on each prescribed fire deployment is shown in Fig. S6. Any OAS that operated without issue is shown in green. OAS that experienced a power failure or other technical failure are shown in red and orange, respectively. Power failure was defined as sampling for less than the 24-hour goal due to depletion of battery power. Other common failure modes were if average OAS flow rate was not within 12.5% of specified flow rate (2 L/min) or if the instrument failed to turn on at the specified time. Power consumed by the OAS is strongly dependent on filter loading, which is a function of the sampled aerosol mass concentration. High filter loadings create greater than normal pressure drops across the OAS filter, forcing the pumps to work harder to maintain flow rate. As a result, the OAS consumes more power, which decreases runtime. Eleven of twelve OAS successfully completed the 24-hour sampling period on 9/15/2016. Five OAS fully completed the 24-hour sampling period on 9/18/2016 while 5 experienced power failure. Depleted from 4 previous 24-hour sampling periods and the high PM$_{2.5}$ concentrations experienced on 9/17/2016, 9/18/2015 saw 5 of the 12 samplers fail due to lack of power.

[Figure]

[Figure]

**Figure S6**. The operational status of each OAS at the conclusion of each sampling day. Colors represent failure mode; numbers in each rectangle represent OAS sampling locations as identified in Figure 3. Asterisks represent high-PM$_{2.5}$ concentration events (leading to premature OAS power failure) that were extrapolated out to 24-hr time-weighted averages for inclusion into Figures 7 and S3. To note, these failure events were not included in Figure 8, which compared valid OAS measurements against reference PM$_{2.5}$ measurements.

[Figure]

**Figure S7.** Time series data for Sharp sensors co-located outdoors in a shaded enclosure for 170 hours (~7 days). For this deployment, one millivolt of sensor output corresponds to approximately 1 $\mu g/m^3$ of $PM_{2.5}$ concentration. Selected 'events' annotated 1-5 are described in the text below and serve as examples of why sensor drift rendered the Sharp data unreliable in the field.

1. Hour 1: At the start of the sampling period, the four Sharp sensors are offset from each other by nearly 100% from lowest to highest, reporting PM concentrations that vary from 12 to 22 $\mu g/m^3$. The lack of precision among co-located sensors is common.

2. Hour 20: The close agreement between Sharp sensors #3 and #4 begins to diverge. Based on examination of all sensor traces, Sharp #3 appears to be experiencing drift relative to the other three devices.

3. Hour 95: All four Sharp sensors experience a sudden rise in output voltage, which produces a dramatic change in both their absolute readings and relative offsets. This effect appears to persist from this point forward. Note how far Sharp 1 is now offset from the other three Sharp sensors.

4. Hour 116: Sharp #2 experiences a phantom spike not seen by the other three sensors (all sensors were co-located in a single box).

5. At the end of the sampling period (hour 170), the four Sharp sensors now report values that are offset from 25 to 65 $\mu g/m^3$ from each other, a factor of four increase in the difference from lowest to highest reading relative to the start of the sampling period.

---

## Author Comment (AC2) · 9 Jan 2018

We thank the reviewer for their insightful comments, which have allowed us to produce a stronger manuscript. Our responses to the general and specific comments are given below. Please note that line number references pertain to the "tracked changes" version of the revised manuscript.

Reviewer 2 general comments

This paper by Kelleher describes the design of a low-cost and field portable PM2.5 monitor that would be of interest to many readers, especially related to remote sampling without access to mains power. The authors give a thorough description of the design, components of their monitor that would enable one to replicate their monitor as well discussion on the consideration that went into the design. The authors demonstrated the use of the monitor in the field, describe the use to spatial mapping the distribution

of smoke from a prescribed burn in Colorado, and demonstrate that the monitor was able to successfully capture daily PM2.5 mass concentrations that compared well to reference instruments. The only downside to the paper is that I would have liked to have seen more discussion on the on-line optical sensor (Sharp) to monitor PM and the reasons why it failed. This will help the reader understand the limitations of the Sharp sensor and so avoid similar problems. In this reviewer opinion, the manuscript falls within the scope of AMT and should be considered for publication after consideration of the minor points below.

Please find our responses to the comments and clarifications detailed below, and details of the corresponding changes in the revised manuscript.

Reviewer 2 specific comments:

1. Comment: Page 4, line 95: Perhaps the authors could outline why they chose the Sharp optical sensor over the myriad of other lowcost particle sensors available commercially.

Response: lines 95-96 have been changed to outline why the Sharp was chosen over other low-cost sensors, "Wang's evaluation of the Sharp demonstrated a linear response with aerosol concentration change and less dependency on atmospheric variables with respect to other low-cost sensors evaluated."

2. Comment: Section 2.4: what sampling height did you place the monitors?

Response: The OAS samplers were placed at a height 1 meter off the ground to prevent sample contamination by ground dust and foliage.

We have revised line 153 of the text to read: "Each OAS was placed on a tripod at a height of 1 m and at a minimum of 60 m from the nearest road to avoid the influence of road dust emissions."

3. Comment: Page 9, line 216: Perhaps you could compare the total battery life that was achieved during the measurements compared to the simulations?

Response: The ratio of energy collected by OAS during field deployment to the 24-hour average of solar irradiance striking the solar panels was 6.7 %. Solar irradiance reaching ground level can be absorbed or reflected by fire emissions overhead, reducing the flux of solar energy available for conversion by OAS solar cells. The 6.7% average efficiency was slightly less than the anticipated 7.5% efficiency used in the Monte Carlo model and design phase. Because aerosol loading effects were a primary determinant of reduced battery life (and because these effects were not modeled in the Monte Carlo simulation), it would be difficult to make a useful comparison of predicted vs. actual battery life.

4. Comment: Page 11, line 240: While I agree that you should correct for collection efficiency of the filter, it would help if the authors were a bit more explicit in how the correction was applied

Response: We apologize for the confusion; the specific methods for estimating filter collection efficiency and applying those corrections to reported data were described in detail in the manuscript appendices but not in the manuscript itself. We have added the following clarification to line 256: "The estimated mass collection efficiency of these filters was 66.7% (see supplemental material for a description of the method to evaluate filter collection efficiency), assuming a size distribution for an unaged biomass burning aerosol (Sakamoto et al., 2016). Mass concentration data reported here have been corrected for filter collection efficiency."

For your reference, we have uploaded the revised appendices with this response.

5. Comment: Page 11, line 245: why would the high mass loading reduce battery power, was it because the pumps had to work harder than expected?

Response: Yes, higher particulate loadings induce higher pressure drops across the filter meaning pumps work harder to maintain a specific flow rate through the filter. Please see our response to Reviewer 1, Comment 2 for more detail.
6. Comment: Figure 7: For each site on the maps you give an 'average' PM2.5 daily mass concentration. How is this an average concentration when each map is one day of measurements and there is presumably one OAS at each site?

Response: The numbers shown in Figure 7 represent 24-hr, time-weighted average concentrations (i.e., gravimetric PM2.5 mass) for a given OAS at each location. The numbers refer to time-averaged concentrations and the colors refer to interpolated values between monitor locations.

7. Comment: Figure 7 and S3: what happed to data from the 11th, 13th, 14th and 16th Sept? Why is data from these days not included in the Supplement?

Response: There was an 8-hour break between each sampling event because of the time required to complete the 102-mile round trip and service all of the samplers (replace filters, extract data, check flow rate, etc). Thus, we did not conduct sampling on all fire days. Fire operations were paused the 13th and 14th due to poor weather conditions and rain.

8. Comment: As mentioned previously, I think more discussion on the why the Sharp optical sensor failed would be useful. The authors mentioned baseline correction was attempted but give no details, what variables were tried but failed? Or to put it another way, how did you come to conclusion that the baseline was not correctable? As the strong linear trend with temperature suggested that it could be correctable. By giving details of what did not work is just as valuable for the reader, so that they can avoid similar issues with this sensor. Furthermore, have other people reported the same problems (e.g. Wang et al. 2015)?

Response: Besides a strong dependence on temperature, the Sharp sensors showed a baseline drift that was difficult to predict. We have created a supplementary Figure (S7) that depicts the various forms of baseline drift among four Sharp sensors that were co-located outdoors for seven consecutive days. These sensors were placed in a box (in the shade) with a common fan circulating air across the devices. The S7 time series

plot has been annotated to specifically highlight issues with sensor drift that rendered the Sharp sensor unreliable.

9. Comment: Figure 9: What about the results from the second location were the OAS was co-located with reference instruments, did it perform just as well?

Response: Results between OAS and reference monitor at location 9 were inconsistent and excluded from our analyses because we do not believe we could achieve an adequate comparison at this site. Two reasons contributed to this decision. First, at the request of the operator, the OAS was located approximately 10 m away from the reference monitor at this site. While a 10m separation distance would normally not be a big issue for a co-location study, the reference instrument was located just a few meters from a gravel road (i.e., a local PM source). This road experienced considerable vehicle traffic during the study (a 50-person camp associated with fire operations was located just meters away). Second, vehicles regularly passed by these locations, generating extremely high dust levels. Because the OAS and reference monitor were not located immediately adjacent to each other (our monitor was farther away from the road) we believe the comparison at this location is not valid.

10. Comment: Page 16, line 333: there have been a few recent papers that have found reliable low-cost and small optical particle counters (e.g. Crilley et al.2017 & Sousan et al. 2016). I think the authors could reference these and other papers here with discussion on whether these other sensors may be useful and practical.

Response: We agree and have added these references to the manuscript, line 380, where we discuss future work to incorporate improved optical sensors into the OAS.

Please also note the supplement to this comment:
https://www.atmos-meas-tech-discuss.net/amt-2017-358/amt-2017-358-AC2-supplement.pdf